# Research on anti-rollover active control of sports utility vehicle with time-delay compensation function

Dongtao Wang[1], Yifan Hu[2], Xuan Liu[1], Yanjian Shen[1], Rong Wang[1]*

**1** Hunan Automotive Engineering Vocational University, Zhuzhou, China, **2** Hunan University of Technology, Zhuzhou, China

* wangrong230116@163.com

## Abstract

The global incidence of traffic accidents caused by vehicle rollovers has exhibited a persistent upward trajectory in recent years. This paper proposes a novel rollover prevention control method incorporating time-delay compensation to address inherent latency issues in anti-rollover control systems (ARCS). First, structural parameters and dynamic theory establish a three-degree-of-freedom (3-DOF) dynamics model for a sport utility vehicle (SUV). Subsequently, a lateral load transfer ratio (*LTR*) estimation model is developed and validated under J-turn test conditions. A grey prediction model is then implemented to forecast *LTR* values in advance, compensating for system time delays. A two-dimensional fuzzy controller, utilizing error and error change rate as inputs, generates corrective yaw moment through differential braking to maintain vehicle stability. Co-simulation experiments conducted in CarSim and MATLAB/Simulink under typical driving scenarios demonstrate that the proposed method effectively mitigates ARCS time delays while preserving driving stability. The results suggest this approach provides both a practical solution for SUV rollover prevention and a conceptual advancement for vehicle active safety systems, showing strong potential for real-world implementation to reduce rollover risks and enhance road safety.

## 1. Introduction

At present, road traffic accidents represent a significant cause of mortality. More than one million people are injured annually by road accidents [1]. The prevention of vehicle rollover is a critical factor in ensuring the safety of drivers and passengers. According to the U.S. National Road Safety Administration, although vehicle rollover accidents account for only 1.7% of all traffic accidents, they have a mortality rate as high as 33% [2]. The propensity for rollover accidents is particularly pronounced in sport utility vehicles (SUVs) due to their higher center of gravity and narrower wheelbase, rendering them more susceptible to rollover when steering to avoid

**Data availability statement:** All relevant data for this study are publicly available from the GitHub repository (https://github.com/wangrong714/anti-rollover-active-control).

**Funding:** This work was supported by the Hunan Provincial Natural Science Foundation (grant number 2023JJ60216, 2023JJ60217), the Hunan Provincial Department of Education Scientific Research Outstanding Youth Project (grant number 24B1026), Humanities and Social Sciences Research Planning Fund of the Ministry of Education (grant number 24YJAZH124).

**Competing interests:** The authors have declared that no competing interests exist.

obstacles or changing lanes. This elevated risk is substantiated by statistics showing that the rollover rate for SUVs is over nine times higher than typical sedans, with a correspondingly higher casualty rate [3]. Consequently, the development of rollover prevention technology has emerged as a prominent research focus within the domain of automobile active safety [4,5]. This paper mainly studies how to compensate for the time-delay phenomenon in the vehicle anti-rollover control system (ARCS) and provides a new method for improving it.

A vehicle ARCS is an effective method of preventing vehicle rollovers. In recent years, numerous experts have expended considerable effort in this domain. Zhao et al. presented a comprehensive anti-rollover control strategy to address the rollover problem of emergency rescue vehicles [6]. Nguyen et al. advanced a novel fuzzy control device for direct active anti-roll bars to enhance the efficacy of vehicle rollover stabilization [7]. Zhu et al. developed a novel ARCS to bolster vehicle steering stability by decreasing sideslip angle and yaw rate [8]. Vu et al. created an active anti-roll bar control technique to improve the vehicle's anti-rollover performance during turning maneuvers [9]. Vu et al. employed a robust control approach to develop a new control technique for an active anti-rollover model. This vehicle's structural design was conceived to enhance its stability in the presence of lateral forces. This goal was achieved by incorporating an additional suspension camber sensor [10]. Guo et al. have proposed a novel active hydraulic system actuator that can mitigate the vehicle roll angle caused by various conditions. This actuator mitigated the risk of vehicle rollover [11]. Wang et al. proposed an integrated control strategy of distributed electric vehicles based on particle filter state estimation to realize the integration of handling, lateral stability, rollover prevention and ride comfort of distributed wheel motor-driven electric vehicles [12]. Jing et al. proposed a new integrated multi-objective control strategy for electric vehicles using multiple actuators to ensure the vehicle's anti-roll and lateral stability [13]. Liu et al. proposed a 3-D Path Planning System Considering rollover and path length, which can plan a shorter path based on suppressing the roll angle of autonomous vehicles [14]. A comprehensive review of these studies reveals that they have yielded promising results in vehicle rollover prevention control. However, a closer examination reveals that their primary objective is to minimize the body roll angle. The discrepancy between the desired rollover assessment index and the actual index is subsequently utilized as the input signal to the controller. Thereby, the anti-rollover process is achieved by controlling the output value. However, the critical point of the rollover angle is relatively ambiguous, as it only reflects the degree of body tilt and does not directly reflect the change of wheel load and the actual rollover risk faced by the vehicle. Consequently, the rollover angle is not a suitable choice. Among the commonly used vehicle rollover evaluation indicators, the lateral load transfer rate (*LTR*) is an indicator that is independent of vehicle parameters and driving conditions [15]. This index has been widely used in the analysis and control of vehicle rollover because of its versatility and ability to reflect the rollover tendency of the vehicle. Nevertheless, given that this index is associated with the vertical load of tires on both sides of the vehicle, its calculation is not possible in a direct manner. It is imperative to estimate the *LTR* accurately.

Additionally, the vehicle's active ARCS is subject to temporal delays in signal collection, signal processing, control algorithm calculation, and actuator initiation. Any one of them will cause the overall delay of the system individually or in combination [16,17]. The presence of this time delay not only impacts the control performance but also contributes to the destabilization of the control system. Consequently, scholars have researched the time delay control of control systems domestically and internationally. Luo et al. constructed a novel Lyapunov-Krasovskii generalized function. They then derived a novel stability criterion for the sampling data control system with multiple time-varying time delays in the form of a linear matrix inequality. This stability condition was derived from the proposed generalized function [18]. Manikandan et al. employed the frequency domain method to ascertain upper limits on the time delay of an aircraft pitch control system, taking into account gain and phase margin constraints. They further examined the efficacy of proportional-integral (PI) and fractional-order PI controllers on the stability and performance of an aircraft pitch control system in the context of a time delay [19]. Liu et al. proposed a robust compensator composed of a nominal and signal-based robust controller for an uncertain quadrotor engine with input delays [20]. Wang et al. advanced a novel proposition for an output feedback controller, one that is contingent upon time delay and quantization density patterns [21]. Tao et al. studied a quantitative iterative learning control based on a coding and decoding mechanism, which effectively solved the problem of data loss in a grid control system when the communication bandwidth was limited, and the load was high [22]. Zhang et al. proposed a prescribed time adaptive dynamic programming control method, which can optimize the steady-state performance of the nonlinear time-varying delay system and ensure that the system can accurately complete the task within a specific time range [23]. Peng et al. designed a fault estimator for spatiotemporal faults and used the iterative learning strategy to estimate the faults accurately [24]. While these methodologies can enhance the control system's resilience to time delays and facilitate its operation in such environments, enhancing system robustness inevitably comes at the expense of control precision. Suppose the control system can predict and control the rollover evaluation index in advance. In that case, it can compensate for the impact of time delay and ensure the control accuracy.

To sum up, most of the existing researches on vehicle rollover prevention methods mainly focus on the control of body roll angle, which is not the best choice among the various indicators of vehicle rollover prevention and does not consider the impact of time delay on the system control performance. In order to solve the limitations of previous studies, first of all, in selecting vehicle anti-rollover indicators, this paper selects *LTR*, which is universal and intuitive, as the vehicle anti-rollover indicator. Then, it establishes and verifies the *LTR* estimation model of the SUV. Then, considering the time-delay problem in the control system, the *LTR* is predicted based on the grey prediction model. Finally, the fuzzy controller is controlled in advance according to the predicted *LTR* to ensure the control performance. Therefore, the control method proposed in this paper can compensate for the time delay of the vehicle's active ARCS, ensure control accuracy, and improve the vehicle's active safety.

## 2. Rollover dynamics model

This paper assumes that the vehicle's longitudinal and pitching dynamics are negligible, neglects the influence of the unsprung mass on the roll dynamics, and presumes symmetry in the dynamic behavior of the left and right tires about the *X*-axis. Therefore, the rollover dynamics model of a 3-DOF SUV, composed of lateral, roll and vertical motions, can be established [25]. The lateral and sideways tilt dynamics models of SUVs are illustrated in Fig 1, respectively.

The differential equations of SUV lateral, roll and vertical motion are as follows [26]:

$$\sum F_y = m A_y = F_{yrl} + F_{yrr} + \left( F_{yfl} + F_{yfr} \right) \cdot \cos \delta \tag{1}$$

$$\left( I_{xx} + mh^2 \right) \cdot \left( \ddot{\varphi} - \ddot{\varphi}_r \right) = (F_{zl} - F_{zr}) \cdot \frac{d}{2} + \sum F_y \cdot h \cdot \cos \varphi + mgh \cdot \sin \varphi \cdot \cos \varphi_r$$
$$-mgh \cos \varphi \cdot \sin \varphi_r + \left( (I_{yy} - I_{zz}) - mh^2 \right) \cdot r^2 \cdot \sin (\varphi - \varphi_r) \cdot \cos (\varphi - \varphi_r) \tag{2}$$

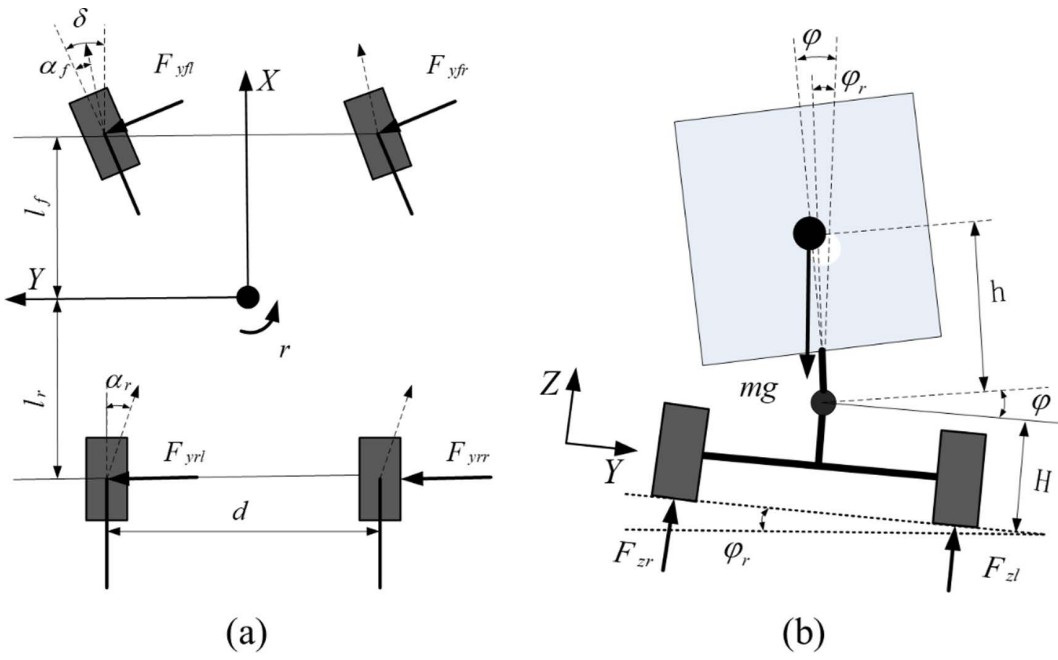

**Fig 1. Vehicle dynamics model: (a) Vehicle lateral dynamics. (b) Vehicle roll dynamics.**

$$m\ddot{z} = m \cdot \left( \dot{\varphi}^2 h \cdot \cos\varphi + \ddot{\varphi} \cdot h \cdot \sin\varphi \right) = (F_{zl} + F_{zr}) - mg \cdot \cos\varphi_r$$

(3)

Where $A_y$ is lateral acceleration, which can be expressed as:

$$A_y = \dot{v} + ru - g\sin\varphi_r + hr^2\sin\varphi + h\dot{\varphi}^2\sin\varphi - h\ddot{\varphi}\cos\varphi$$

Assuming that the inclination angle of the road surface is 0°, Eqs from (1) to (3) can be expressed as:

$$\sum F_y = m \cdot \left( \dot{v} + ru + hr^2 \cdot \sin\varphi + h\dot{\varphi}^2 \cdot \sin\varphi - h\ddot{\varphi} \cdot \cos\varphi \right)$$

(4)

$$\left( I_{xx} + mh^2 \right) \cdot \ddot{\varphi} = (F_{zl} - F_{zr}) \cdot \frac{d}{2} + \sum F_y \cdot h\cos\varphi + mgh\sin\varphi$$
$$+ \left( (I_{yy} - I_{zz}) - mh^2 \right) \cdot r^2 \cdot \sin\varphi \cdot \cos\varphi$$

(5)

$$m\ddot{z} = m \cdot \left( \dot{\varphi}^2 h \cdot \cos\varphi + \ddot{\varphi} \cdot h \cdot \sin\varphi \right) = (F_{zl} + F_{zr}) - mg$$

(6)

## 3. Estimation of *LTR* of SUV

During a high-speed corner, the vehicle's vertical load is transferred laterally to the tires outside the corner. In contrast, the load on the tires inside the corner decreases, and the vehicle tilts to the outside of the corner. In general, the load of the

internal test tire can be reduced to zero. When the tire is about to leave the ground, it can be used as the boundary point to judge the SUV rollover. The *LTR* is frequently utilized as an indicator of the vehicle's proximity to be upside down [27], and The calculation formula is as follows:

$$LTR = \frac{F_{zr} - F_{zl}}{F_{zr} + F_{zl}}$$

(7)

Eq (7) implies that *LTR* is a unified index with a value range of [-1,1]. Assuming the barycentre of the SUV is on the longitudinal axis, when the SUV is not rolling, the vertical load on both tires is equal, $V=0$. When the SUV reaches the critical point of rollover, that is, when one tire leaves the ground, $V=\pm1$. Since this index only considers the stress on the tires, the degree of rollover can be uniquely determined for different models and driving conditions. Therefore, using this index to indicate the degree to which a vehicle is about to roll over is common.

It is acknowledged that measuring the vertical load of tires on both sides while the vehicle is in operation presents a significant challenge. This is because the *LTR* cannot be calculated directly using Eq (7). Consequently, a commonly employed method to estimate the *LTR* is to utilize easily measurable vehicle state quantities [20]. It is assumed that the SUV roll angle changes slowly, which is $\ddot{\varphi} \approx 0$ and $\dot{\varphi} \approx 0$, and $\cos^2\varphi \approx 1$, $r^2 \approx 0$. Plugging Eq (5) and Eq (6) into Eq(7) can get:

$$LTR_e = \frac{2}{d} \cdot \frac{h \cdot (\cos\varphi \cdot (\dot{v} + ru) + g \cdot \sin\varphi)}{g}$$

(8)

Substituting $\dot{v} + ru = A_y \cdot \cos\varphi$ into Eq (8) and taking the roll angle as the measured value, the estimation model of *LTR* can be obtained as:

$$LTR_e = \frac{2h}{d \cdot g} \cdot \left( A_y' + g \cdot \sin\varphi' \right)$$

(9)

The measured values of lateral acceleration and body roll angle are indicated by $A_y'$ and $\varphi'$, respectively. These are two vehicle condition variables that are easy to measure.

To evaluate the accuracy of the SUV *LTR* estimation model expressed in Eq (9), a dynamic model of an SUV was established using the commercial software CarSim. Vehicle dynamics simulations can precisely measure a vertical load of tires on both sides of the SUV. Consequently, the *LTR* calculation model presented in Eq (7) was developed using Matlab/Simulink and designated as the actual value. The *LTR* estimation model described in Eq (9) was also implemented in Matlab/Simulink and defined as the estimated value. The accuracy of the SUV *LTR* estimation model can be assessed by comparing the actual value derived from Eq (7) with the estimated value obtained from Eq (9). Table 1 lists the parameters of the SUV involved in the *LTR* estimation model, which were extracted from the SUV model in CarSim. The simulation test condition employed is the J-Turn test.

In the J-Turn test, the SUV's initial speed was 110 km/h, the road surface's adhesion coefficient was 0.85, and the steering wheel's maximum angle was 180°. Fig 2 shows the input of the SUV's steering wheel angle. As indicated in Fig 3, the figure elucidates the alteration in *LTR* in the J-Turn test and juxtaposes the estimated value with the actual value.

As evidenced in Fig 3, the estimated *LTR* in the test can accurately track the actual figure with a relative error of only 2.7%, indicating that the actual figure calculated by Eq (7) can be accurately estimated using Eq (9). In other words, the estimation model represented by Eq (9) can accurately estimate the SUV's *LTR*.

**Table 1. SUV numerical simulation parameters.**

| Parameter | Symbol | Unit | Numerical value |
|---|---|---|---|
| Sprung mass | $m$ | kg | 1831 |
| Distance from center of gravity of sprung mass to center of roll | $h$ | m | 0.4 |
| Wheelbase | d | m | 1.55 |
| Body yaw moment of inertia | $I_{xx}$ | kg·m$^{-2}$ | 708.22 |
| Body roll moment of inertia | $I_{yy}$ | kg·m$^{-2}$ | 4521 |
| Body pitch moment of inertia | $I_{zz}$ | kg·m$^{-2}$ | 4607 |

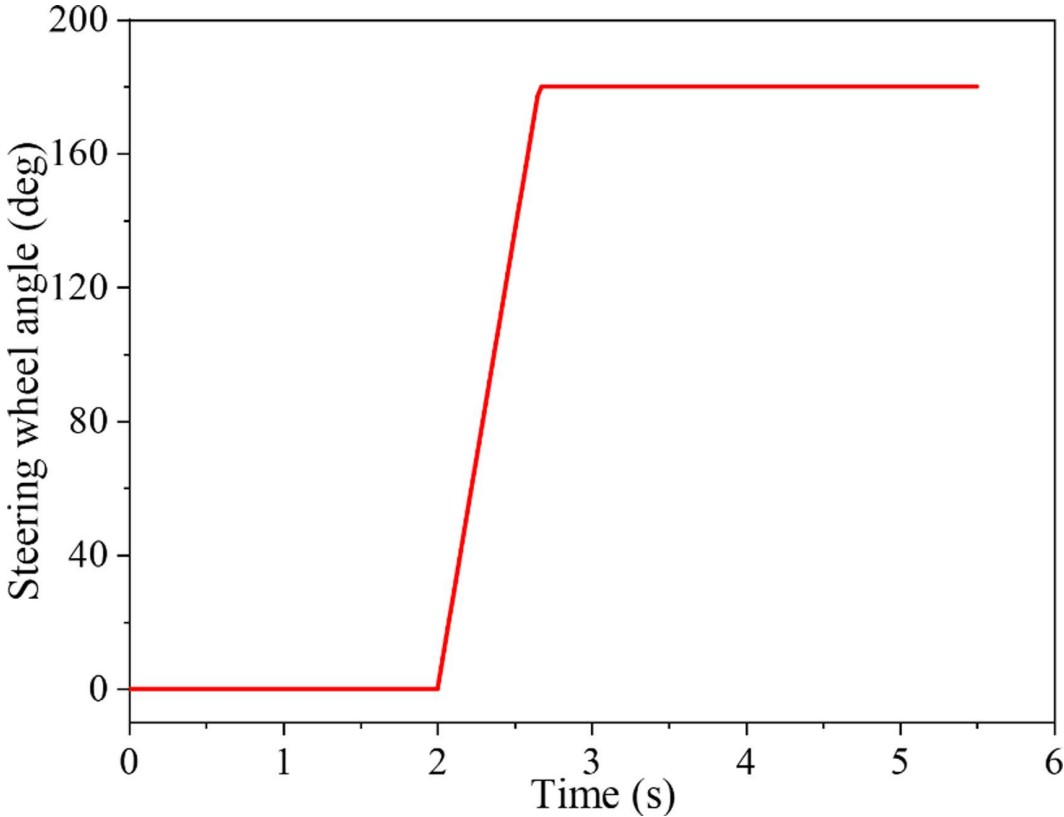

**Fig 2. Steering wheel angle input in J-Turn test.**

## 4. Grey prediction of *LTR* for SUV

### 4.1 Grey prediction

Grey prediction is a scientific and quantitative method for forecasting the future condition of a system. It is rooted in the principles of grey theory and involves sorting and analyzing original data to identify patterns that indicate change. It involves organizing original data and identifying its changing patterns. The method mainly includes recognizing the characteristics of system evolution uncertainty, using sequential operators to generate and process original data, mining system evolution laws, and building a grey system model [28]. Grey prediction is a member of the family of nonlinear

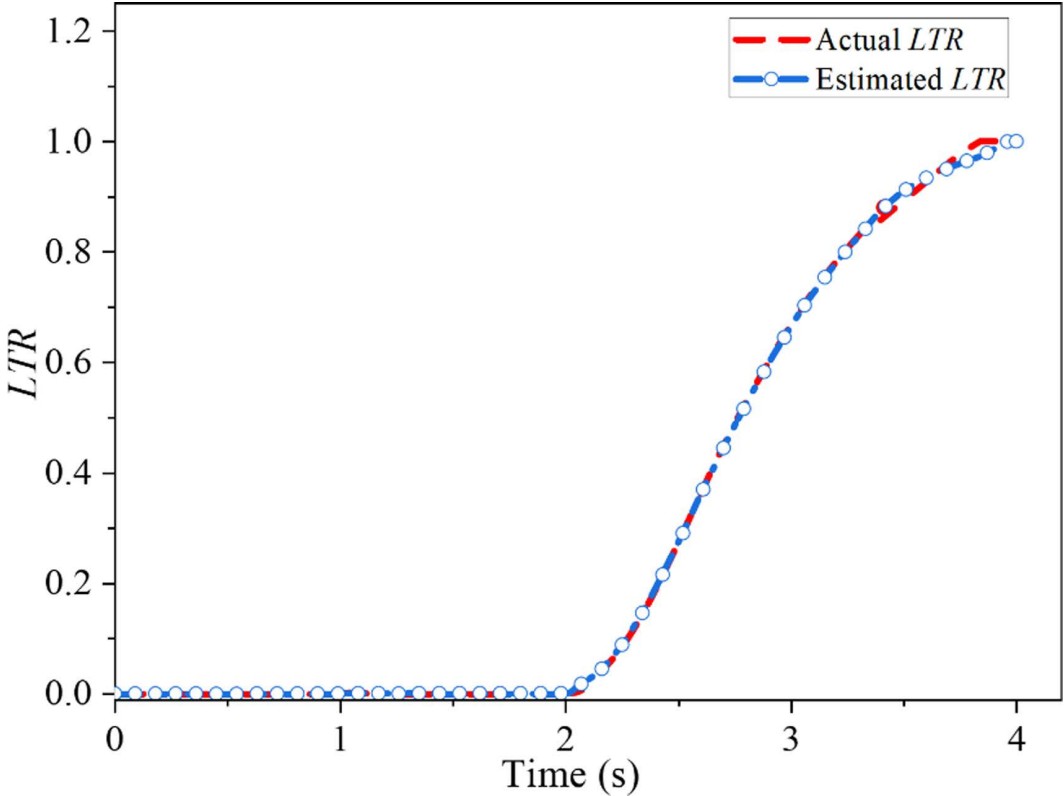

**Fig 3. Comparison of estimated value and actual value of *LTR* in J-Turn test.**

extrapolation prediction methods. These methodologies have been extensively employed in numerous social and natural sciences research endeavors. The primary rationale for their extensive use is their capacity to analyze and model systems with short time series, limited statistics, and incomplete information [29–31].

Grey prediction typically employs the GM (1, 1) model [32], which predicts founded on the actual input discrete values of the prediction model. This model requires the identification of only two parameters (development coefficients $\hat{a}$ and grey effects $\hat{u}$) to predict the process with a certain trend over a brief period, and its prediction is accurate. The GM (1,1) model calculation process is outlined below:

**Step 1:** Generate the original input data array of the prediction model.

$$X^{(0)} = \left[ x^{(0)}(1), x^{(0)}(2), x^{(0)}(3) \ldots, x^{(0)}(n-1), x^{(0)}(n) \right]$$

(10)

Where $n$ is the modeling dimension, indicating the amount of historical data used in forecasting. $x^{(0)}(i), i = 1, 2, 3, 4, \cdots, n$ is the input of the prediction model.

**Step 2:** Data preprocessing

The raw data collected in the system may contain positive and negative numbers. The direct addition of these numbers can result in information loss. This phenomenon can occur when positive and negative numbers are added

together. As a result, the regularity of the series generated by the addition operation may be weakened or even eliminated. Therefore, it is necessary to map the original data to a non-negative series before adding it.

$$X_m^{(0)} = \left[ x_m^{(0)}(1), x_m^{(0)}(2), x_m^{(0)}(3) \ldots, x_m^{(n-1)}, x_m^{(0)}(n) \right]$$

(11)

In this paper, exponential mapping is used to preprocess the original data, that is, $X_m^{(0)}(k) = \exp[r \times x^{(0)}(n)]$, where $r$ transforms coefficients exponentially.

**Step 3:** An accumulation of preprocessed raw data can be obtained.

$$X_m^{(1)} = \left[ x_m^{(1)}(1), x_m^{(1)}(2), x_m^{(1)}(3) \ldots, x_m^{(1)}(n-1), x_m^{(1)}(n) \right]$$

(12)

Where $x_m^{(1)}(k) = \sum_{i=1}^{k} x_m^{(0)}(i) \; ( \quad k = 1, \; 2, \; \ldots, \; n.$

**Step 4:** Make up the data matrices $B$ and $Y$.

$$B = \begin{bmatrix} -0.5 \left( x_m^{(1)}(2) + x_m^{(1)}(1) \right) & 1 \\ -0.5 \left( x_m^{(1)}(3) + x_m^{(1)}(2) \right) & 1 \\ \cdots & 1 \\ -0.5 \left( x_m^{(1)}(n) + x_m^{(1)}(n-1) \right) & 1 \end{bmatrix}, \; Y = \begin{bmatrix} x_m^{(0)}(2) \\ x_m^{(0)}(3) \\ \cdots \\ x_m^{(0)}(n) \end{bmatrix} \cdots$$

(13)

**Step 5:** Calculate the development coefficients $\hat{a}$ and $\hat{u}$.

$$[\hat{a}, \; \hat{u}] = \left( B^T B \right)^{-1} B^T Y$$

(14)

**Step 6:** According to the model GM (1,1), the predicted value $\hat{x}^{(0)}(j+k)$ at time $j$ to time $j+k$ is expressed as:

$$\hat{x}_m^{(0)}(j+k) = \left( \hat{x}_m^{(0)}(j-n+1) - \hat{u}/\hat{a} \right) \cdot \left( \exp\left( -\hat{a}(k+n-1) \right) - \exp\left( -\hat{a}(k+n-2) \right) \right)$$

(15)

Where $\hat{x}_m^{(0)}(j-n+1)$ is the predicted value at $j$-$n$+1 moment. $k$ is the number of steps predicted in advance.

**Step 7:** It is also necessary to perform inverse mapping on the data to obtain the original forecast data, that is

$$\hat{x}^{(0)}(j+k) = \frac{1}{r} \ln \left[ \hat{x}_m^{(0)}(j+k) \right]$$

(16)

The precision of the model's predictions is contingent not solely on the distribution of the original data series but also on the dimensionality of the model itself [33,34]. An overabundance of modeling dimensions, characterized by an excess of old information that overwhelms new information, can impede the model's responsiveness to system fluctuations, deteriorate the tracking performance, and diminish the computation speed. The determination of the optimal modeling dimension can be achieved through a process of trial and error.

## 4.2 Prediction of the LTR and analysis of its accuracy

Through literature review and technical consultation, it is evident that the time delay of the vehicle ARCS typically ranges from 0 to 0.2s [35]. It has been established that the longer the prediction time, the lower the forecast precision. Therefore, it is imperative to analyze this model's precision thoroughly in the event of a time delay.

Fig 4 indicates the distinction between the estimated value and the predicted value of the *LTR* within 0.2s of the predicted time under the J-Turn test condition of the SUV. According to this model, the advanced forecast time is proportional to the acquisition time and the advanced forecast steps and has nothing to do with the modeling dimension. When the sampling time $T_s$ is 0.02s, the number of advance prediction steps $k$ is 2, 4, 6, 8 and 10, respectively, which means that the advance prediction time $T$ ($T = T_s \times k$) is 0.04, 0.08, 0.12, 0.16 and 0.2s respectively. As displayed in Fig 4, the forecast curve fluctuation increases and the forecast accuracy decreases as the prediction time increases. However, the predicted value of the *LTR* can maintain the same shape as the estimated value within 0.2s of the prediction time. These indicate that the grey prediction model possesses a satisfactory prediction ability. In order to analyze the prediction accuracy further, the low threshold may lead to the premature start of the system and increase energy consumption. A higher threshold may cause the system to respond too late and increase the risk of rollover. Through experimental analysis and comparison of the results of various data in 0.7~0.9, it can be selected that the control system takes the *LTR* equal to 0.85 as the control threshold, that is, when the *LTR* reaches 0.85, the ARCS of SUV will start to work. By the prevailing theoretical framework, the estimated value of the *LTR* is predicted to reach 0.85 at 3.34s. With an advance prediction time of 0.04s, the *LTR* predicted by the grey prediction model is expected to reach 0.85 at 3.3s. Similarly, if the advance prediction time is 0.08s, the *LTR* value predicted by the grey prediction model should reach 0.85 at 3.26s. However, a discrepancy often emerges between the anticipated and actual values due to the inherent inaccuracy of grey prediction

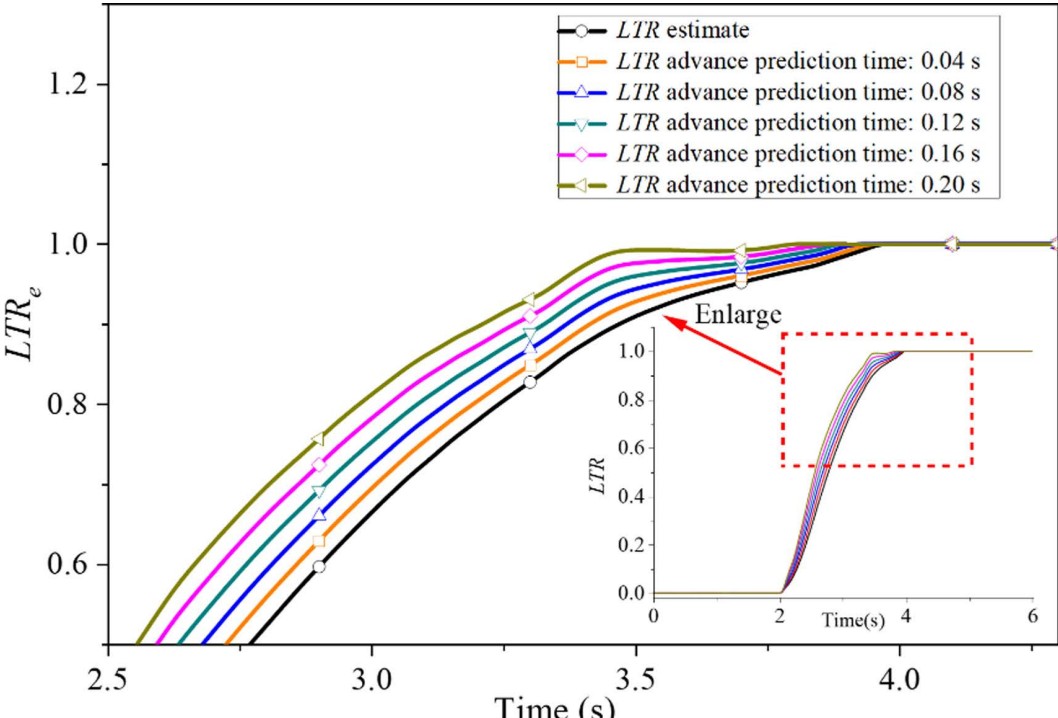

**Fig 4. Effect of advanced prediction time on prediction accuracy.**

**Table 2. Analysis of the error in predicting the *LTR* of moments in advance.**

| Predict time in advance/s | LTR | | |
| --- | --- | --- | --- |
| | Theoretical value | Actual value | Relative error/% |
| 0.04 | 0.850 | 0.852 | 0.2 |
| 0.08 | 0.850 | 0.856 | 0.7 |
| 0.12 | 0.850 | 0.861 | 1.3 |
| 0.16 | 0.850 | 0.869 | 2.2 |
| 0.20 | 0.850 | 0.878 | 3.3 |

methods. Table 2 compares the theoretical and predicted values of the *LTR* for five different advance prediction times. The maximum relative error of the predicted value of the *LTR* at the five prediction moments is about 3%, as shown in Table 2. The results predicted by the grey prediction model may be quite accurate. They could be used as the input signal to compensate for the time delay of the SUV's ARCS. As illustrated in Fig 4 and substantiated in Table 2, the magnitude of the prediction error is found to rise in proportion to the advancement of the forecast time. When the advanced forecast time is 0.04s, the prediction accuracy is the highest when the prediction is two steps ahead. While the prediction time in advance is 0.2s, the error reaches the maximum when the prediction is 10 steps in advance.

Since grey prediction is based on the inherent laws of the original data to predict the future system state, the dimension of the original data influences the emergence of these laws. Therefore, the grey prediction model's modeling dimension also affects the predictions' accuracy. Fig 5 compares the predicted and estimated values of the lateral transfer rate with different modeling dimensions when the advance prediction time is 0.2s. Since the difference between the predicted value and the estimated value of *LTR* with modeling dimension outside [4,10] is too large, the accuracy of the modeling dimension in [4,10] is only analyzed in detail below.

As illustrated in Fig 5, the grey prediction curve with modeling dimension m = 10 exhibits the greatest proximity to the estimate of the *LTR*. With the increase of modeling dimension, that is, more original data are used for prediction, the influence of new information on prediction data decreases, the fluctuation of the prediction curve decreases, and the prediction accuracy increases, but the calculation time increases. Conversely, the modeling dimension decreases, although the calculation time decreases, the prediction data is affected by the change of new information, the fluctuation of the prediction curve increases, and the prediction accuracy worsens. The model dimensions' prediction accuracy and computational speed should be considered in specific applications. In addition, the characteristics of the data to be predicted need to be considered. For slow changes in the system output, it is preferable to use larger model dimensions to enhance the stability of the prediction. Conversely, for system outputs with large fluctuations, it is more appropriate to use smaller model dimensions to improve the predictability of the system.

The prediction and sampling time significantly impact the model accuracy, while the control threshold selection directly affects the system's response time and safety. Therefore, the sensitivity of these variables should be carefully considered when designing and optimizing the ARCS. Changes in these variables will directly affect the ARCS's performance.

Through iterative experiments, this paper determines the optimal modeling dimension (m) and prediction step (k). Specifically, the higher modeling dimension improves prediction stability but reduces responsiveness to sudden system changes. On the contrary, a smaller prediction step (e.g., *k* = 2) enhances the prediction accuracy but cannot fully compensate for the longer time delay. Therefore, the trade-off between accuracy and responsiveness is considered when selecting the best parameters. The most important thing about ARCS proposed in this paper is grey prediction, and the most important thing about grey prediction is the modeling dimension and prediction step. Selecting the best modeling dimension and prediction step can make the performance of ARCS the best.

 

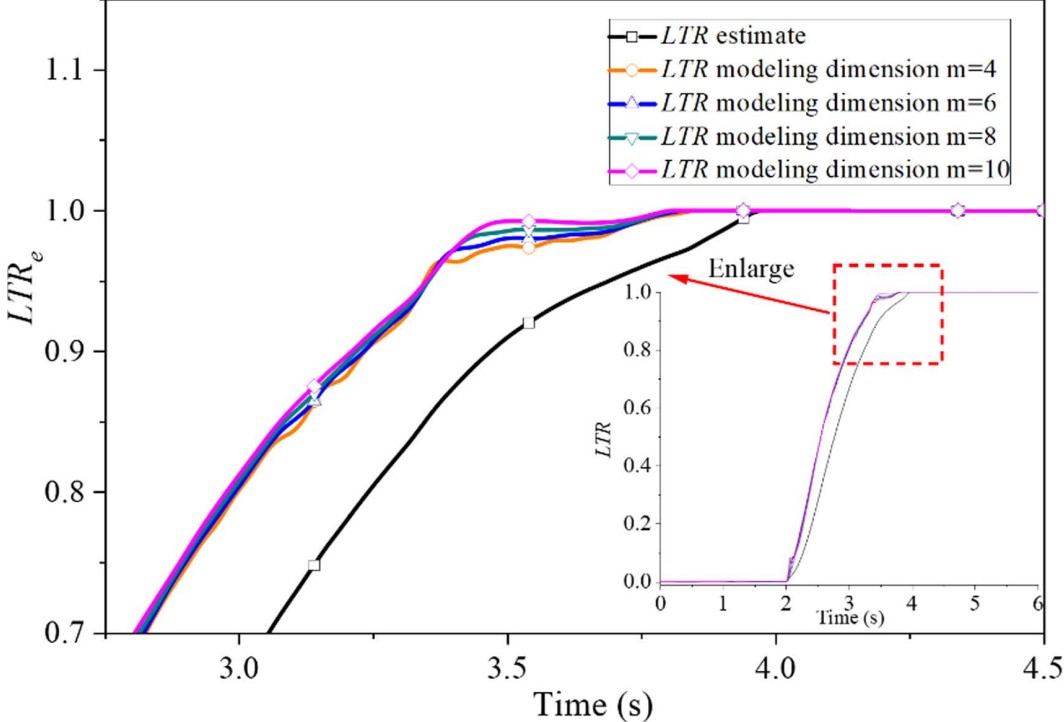

**Fig 5. The effect of modeling dimension on prediction accuracy.**

## 5. Active rollover control system design with time delay compensation

Due to the time delay in the active ARCS of the SUV, the *LTR* representing the rollover risk of the SUV can be predicted in advance by the grey prediction model to compensate. This prediction can serve as the decision input signal for the rollover controller, enabling advanced control. In the specific implementation, the *LTR* estimate of the SUV is first used as the starting value of the model. The time delay of the ARCS determines the time to be predicted in advance, and finally, the *LTR* at a future time is outputted by the grey prediction model. In keeping with the safety of vehicle travel, the *LTR* prediction threshold is set at 0.8, and this is used as the active anti-rollover control trigger condition, i.e., if the predicted value of the *LTR* is greater than 0.8, it is considered that there is a risk of vehicle rollover. The active ARCS must be triggered to perform control. As illustrated in Fig 6, the control flow chart is presented.

### 5.1 Fuzzy controller

Fuzzy control is an intelligent control method predicated on fuzzy set theory, linguistic variables, and fuzzy control logic reasoning. This method simulates the human way of thinking in terms of behavior to implement fuzzy reasoning and decision-making on objects that are difficult to model. Indeed, it can be considered a nonlinear control [36]. This kind of controller design is independent of the accurate mathematical model of the system and has strong robustness, so it is widely used in industrial control processes. These characteristics have led to its wide use in industrial control processes [37,38]. The grey prediction model is based on the discrete value of the actual input. Because of this characteristic, it is very hard to obtain the predicted *LTR* with an accurate mathematical model. Therefore, this paper chooses fuzzy control to design the turning controller.

This paper establishes a decision fuzzy controller for additional yaw moment, which adopts a two-dimensional fuzzy control structure with double inputs and single outputs. The fuzzy controller selects the error *E* between the predicted

value of the transverse load transfer rate and its reference and the rate of change of the error *EC* as the control inputs and uses the additional pendulum moment *M* as the output of the controller to realize the fuzzy controller design, in which the reference value of the transverse load transfer rate is taken to be zero. Seven fuzzy language subsets described both input and output, and the fuzzy sets were: {negative big, negative medium, negative small, zero, positive small, positive medium, positive big} = {*NB*, *NM*, *NS*, *ZO*, *PS*, *PM*, *PB*}. The input and output variables discourse universe was {-6, 6}. The functions belonging to the fuzzy subsets all adopted Gaussian distribution functions. This kind of membership function was not only simple to calculate but also conformed to the characteristics of normal distribution and could satisfy the necessity of control precision.

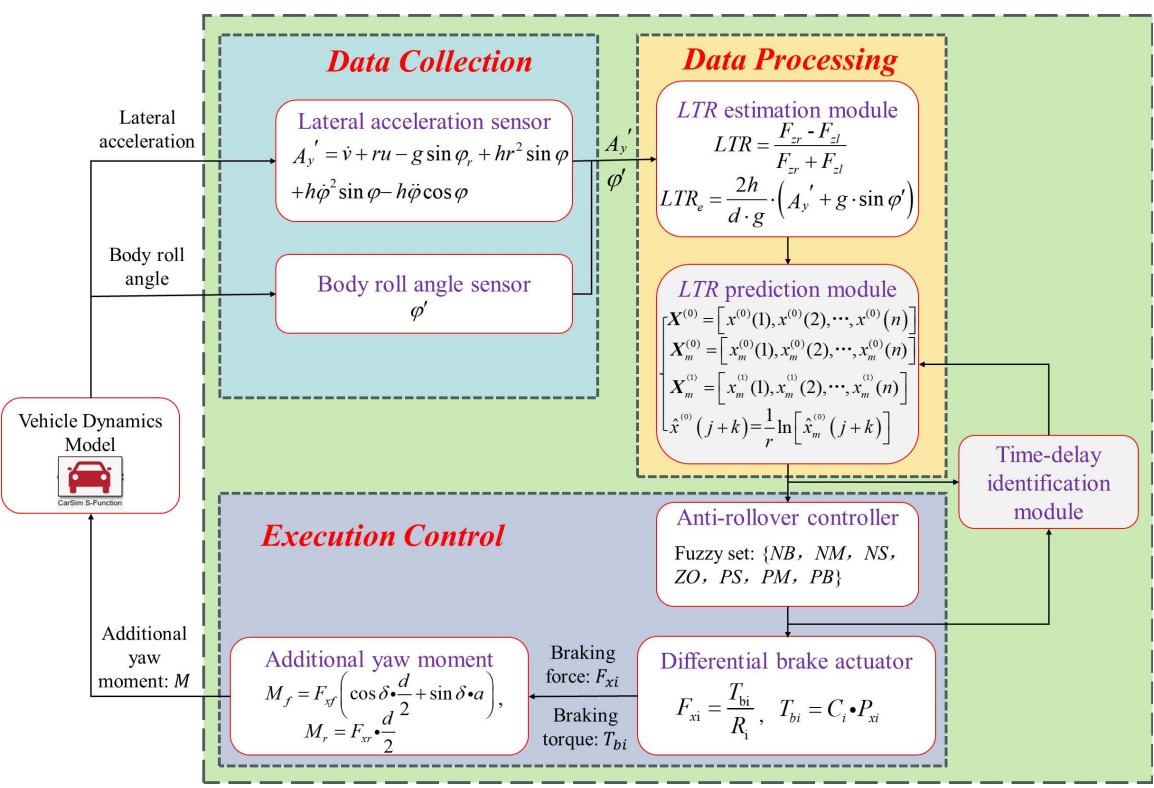

**Fig 6. Block diagram of SUV active ARCS with time delay compensation function.**

**Table 3. Fuzzy control rule table.**

| M | | NB | NM | NS | ZO | PS | PM | PB |
|---|---|---|---|---|---|---|---|---|
| | | | | | **E** | | | |
| | **NB** | PB | PB | PB | PM | PS | ZO | ZO |
| | **NM** | PB | PB | PM | PS | ZO | ZO | NS |
| | **NS** | PB | PM | PM | PS | ZO | NS | NM |
| **EC** | **ZO** | PB | PM | PS | ZO | NS | NM | NB |
| | **PS** | PM | PS | ZO | NS | NM | NM | NB |
| | **PM** | PS | ZO | ZO | NS | NM | NB | NB |
| | **PB** | ZO | ZO | NS | NM | NS | NB | NB |

After determining the input and output variables' fuzzy subsets and membership functions, a table of fuzzy control rules can be prepared, as shown in Table 3. Each fuzzy rule can clearly express a control strategy. For example, when the error between the predicted value of the *LTR* and its reference quantity is negatively large (*NB*) if the rate of change of the error at this time is also positively large (*NB*), it indicates that the error is getting larger. In order to reduce the error as soon as possible, or even disappear, the additional pendulum moment (*PB*) should be increased to achieve the compensation and adjustment of the error to make the vehicle converge to the target value, thus preventing the vehicle from overturning. In order to make the error decrease or even disappear as soon as possible, the additional pendulum moment (*PB*) should be increased to achieve the compensation and adjustment of the error so that the predicted value of the *LTR* is as close as possible to the reference value, thus preventing the vehicle from overturning. If the error change is negative small (*NS*), the error has been gradually reduced, and the control volume (*NM*) should be reduced now. When the error decreases sharply, to prevent over-adjustment, it is better not to change the control amount at this time; that is, at this time, the additional swing moment should be *ZO*, and so on; each fuzzy rule represents a control rule under one situation.

In summary, determining the controller's control rules should adhere to the following principles: if the error is large, the main task of the control quantity is to eliminate the error as quickly as possible. If the error is small, not only should the error be eliminated, but the system's stability should also be considered to prevent the system from unnecessary overshoot and oscillation.

## 5.2 Differential braking control strategy

Differential braking constitutes a control system that engages the brakes on the appropriate tires when the vehicle exhibits instability. By generating additional yaw moments, the vehicle's attitude can be corrected, and the vehicle's instability can be reduced [39]. The additional yaw moment generated by the individual braking of each wheel is different according to the wheel's position. For the front wheel:

$$M_f = F_{xf} \left( \cos \delta \cdot \frac{d}{2} + \sin \delta \cdot a \right)$$

(17)

For the rear wheels, there are:

$$M_r = F_{xr} \cdot \frac{d}{2}$$

(18)

The additional yaw moment is generated by the braking torque produced when the brake wheel applies braking force, resulting from the hydraulic pressure exerted by the wheel cylinder on the brake. The wheel cylinder pressure is calculated using Eqs (19) and (20).

$$F_{xi} = \frac{T_{bi}}{R_i}$$

(19)

$$T_{bi} = C_i \cdot P_{xi}$$

(20)

**Table 4. Tire braking force coordination strategy.**

|  | Turn left | Turn right |
| --- | --- | --- |
| Understeer | Left rear wheel | Right rear wheel |
| Oversteer | Right front wheel | Left front wheel |

Table 4 delineates the tire braking force coordination strategy. In circumstances where the SUV exhibits oversteer during operation, the control system implements braking pressure on the outer front wheels to generate a swinging moment that is antithetical to the steering direction of the SUV, thereby mitigating the oversteer. Conversely, when the SUV exhibits understeer, the control system implements braking pressure on the inner rear wheels. These counteract the understeer and maintain the SUV's driving stability.

## 6. Control performance analysis

In order to verify the effectiveness of the above method, this paper uses Matlab/Simulink and CarSim to conduct simulation experiments. There are many test conditions to choose from, and this paper selects the two most convincing conditions for simulation: step steering [40] and fishhook steering [41]. Then, the stability of vehicle operation without control, without predictive control and with predictive control was compared under two operating conditions.

### 6.1 Step steering condition

The basic parameters under step steering conditions include a starting speed of 100km/h, steering wheel angle of 180°, tire-road friction coefficient of 0.85, and SUV input's front wheel angle, shown in Fig 7. The vehicle enters the circular track after 2s from the straight track. The angle changes rapidly from zero to a fixed value in a short time.

The changes in vehicle evaluation indexes under step steering conditions are revealed in Fig 8. Among them, Fig 8(a) shows the changes in the SUV's center of mass slip angle. The vehicle starts to steer around 2s, the SUV's center of

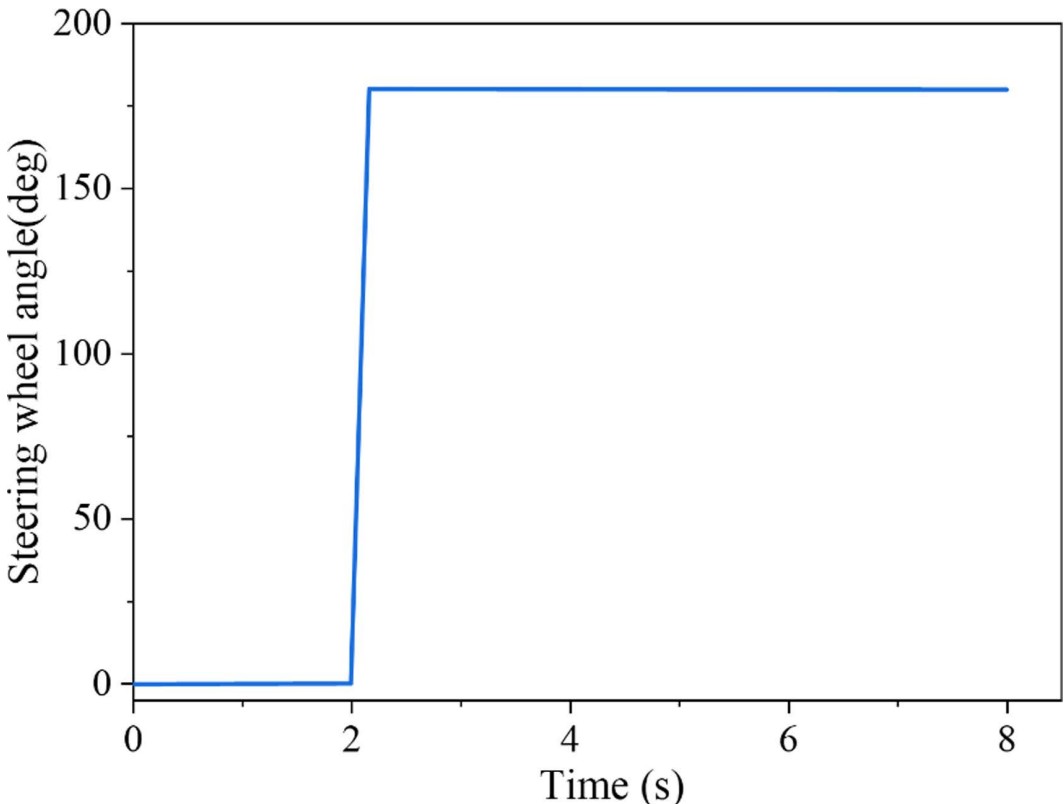

**Fig 7. Steering wheel angle input in step steering condition.**

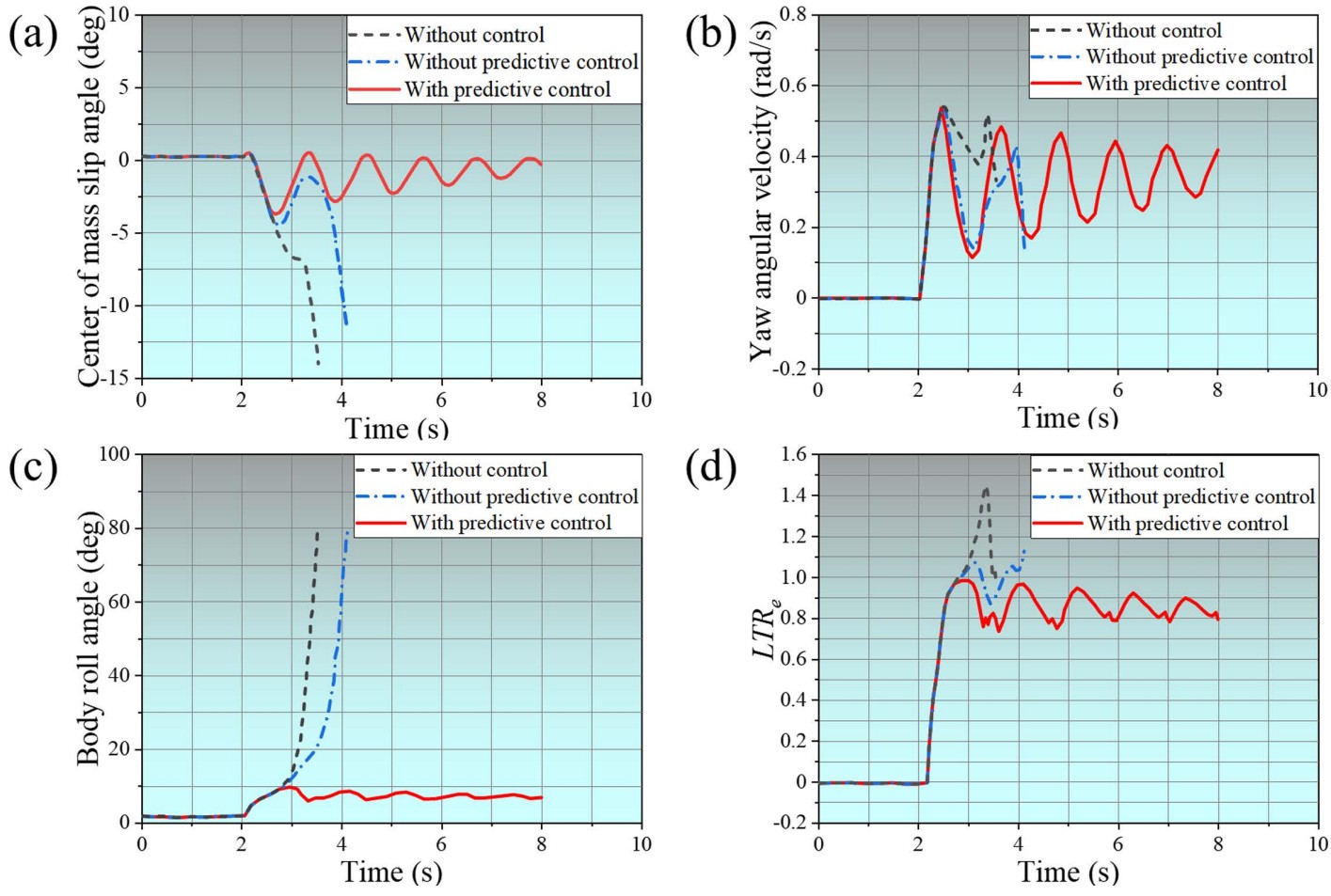

**Fig 8. Changes to Vehicle Rating Indicators.**

mass slip angle without control increases sharply, stops around 3.6s, and the vehicle rolls over. Without predictive control, the SUV's center of mass slip angle decreases around 2.8s, but it increases sharply around 3.5s and stops around 4.2s, and the vehicle rolls over at this time. With predictive control, the SUV's center of mass slip angle gradually decreases during the oscillating change until it approaches zero. Fig 8(b) shows the SUV's yaw angular velocity changes. The yaw angular velocity fluctuates greatly in the two cases of without control and without predictive control; the former stops changing at 3.6s, and the latter stops changing at 4.2s. With predictive control, the SUV's yaw angular velocity gradually stabilizes. Fig 8(c) shows the evolution of the body roll angle. When the vehicle is without control and without predictive control, the body roll angle increases sharply after 2.8s. With predictive control, the body roll angle increased but decreased after 2.8s and gradually stabilized. Fig 8(d) shows the change of the *LTR*, which increases to more than 1 after 2.8s in both cases without control and without predictive control. With predictive control, the *LTR* of the vehicle no longer increased after 2.8s and stabilized at about 0.85.

The phase transition diagram for $\lambda$-$\beta$ and $\beta$-$\dot{\beta}$ are shown in Fig 9. As can be seen from Fig 9 (a), the phase trajectory of the vehicle without control shows a lack of convergence and diverges directly, indicating that the vehicle has rolled over. The phase trajectory of the vehicle without predictive control converges briefly. Eventually, it diverges, indicating that the actuator action lags when the vehicle turns due to time delay, which eventually causes the vehicle to roll over. The phase

trajectory of the vehicle with predictive control converges quickly, indicating that the vehicle operation tends to be stable. From Fig 9(b), it can be seen it can be seen that the phase trajectories of the vehicles concerned fail to converge in both cases without control and without predictive control, resulting in vehicle overturning. Whereas, in the case with predictive control, the vehicles' phase trajectories converge rapidly, indicating that the proposed control method with time delay compensation can make the vehicles turn smoothly.

Comprehensive Fig 8 and Fig 9 show that under the step steering condition, the vehicle rollover time is about 3.6s when the vehicle is without control, and the vehicle rollover time is about 4.2s when the vehicle is without predictive control. The vehicle can complete the steering smoothly when the vehicle is with predictive control. When the vehicle turned without control, all evaluation indicators were unstable, indicating that the vehicle would reverse soon after turning. When the vehicle turned without predictive control, although all evaluation indicators were better than without control, the vehicle still eventually rolled over due to the time delay in the controller's response. However, with predictive control, the grey prediction system can accurately predict the vehicle's driving status and potential risks, pre-calculate the possible time delay problem, and take corresponding compensation measures in time. In this way, the vehicle will trigger the ARCS when it is steering, and the system can carry out the control operation quickly and effectively through advanced preparation and precise prediction. It can accurately adjust the distribution of the vehicle's center of gravity, optimize the contact state between the tires and the ground, and ensure that the vehicle is always stable during the steering process, thus enabling the vehicle to safely and smoothly complete the steering action, effectively avoiding traffic accidents triggered by rollover and other dangerous situations, and safeguarding the safety of the vehicle and the passengers.

## 6.2 Fishhook steering condition

Under the condition of fishhook steering, the vehicle parameters are set as follows: initial speed 80km/h, tire-road friction coefficient 0.85, steering wheel angle change range [-170°, +170°], and the angle input is revealed in Fig 10.

The changes in evaluation indexes when the vehicle turns with a fishhook are displayed in Fig 11. From Fig 11(a), it can be seen that without control, the side slip angle of the center of mass of the SUV changes sharply from around -8 to nearly 20 after 2.2s and stops at 3.4s, indicating that the vehicle is upside down. Without predictive control, the slip angle of the SUV's center of gravity changes steadily until about 3.4s, but after 3.4s, the SUV's slip angle of the center of gravity

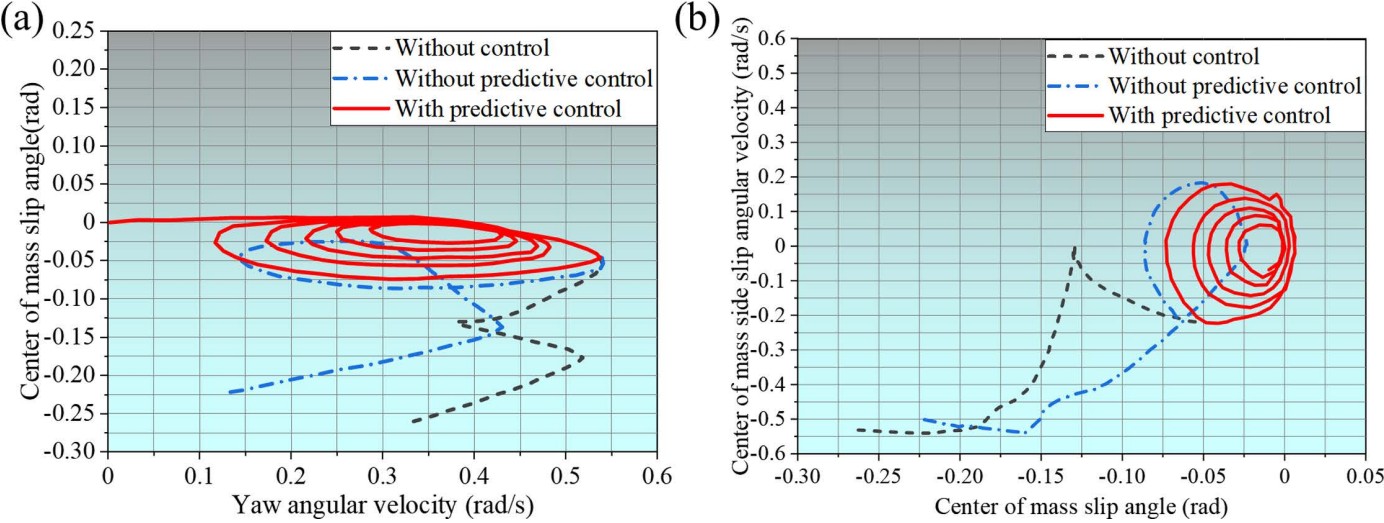

**Fig 9. Phase transition diagram.**

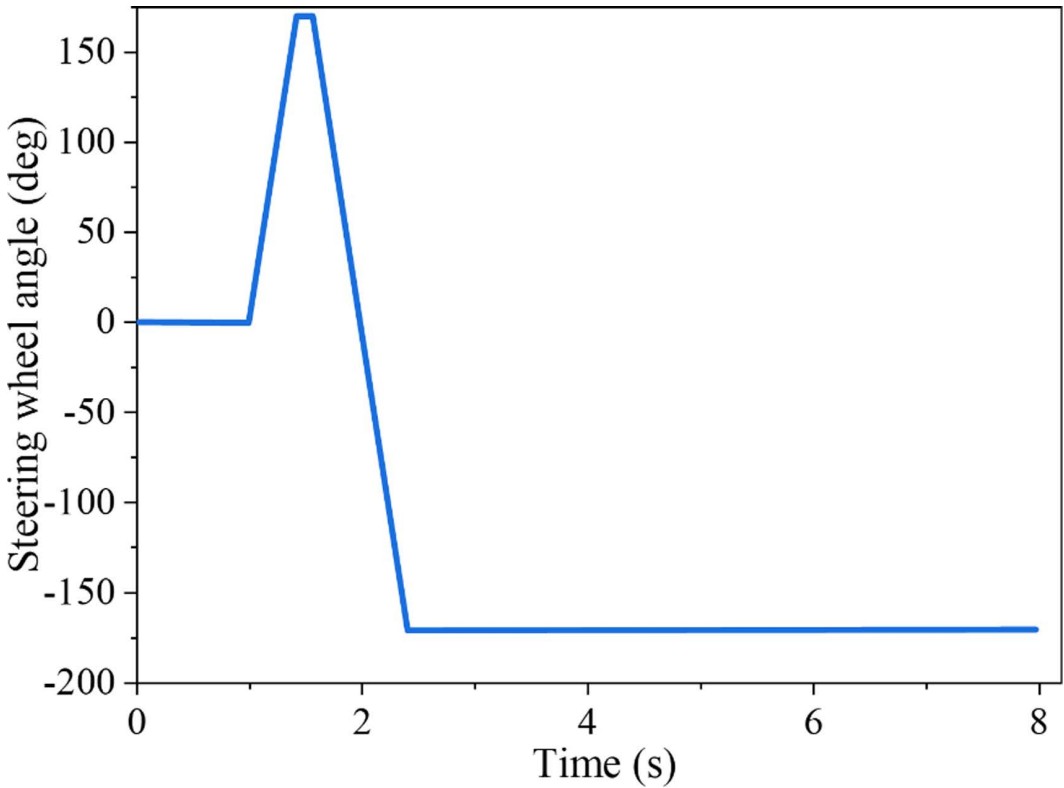

**Fig 10. Steering wheel angle input in fishhook steering condition.**

changes drastically. With predictive control, the change in the SUV's center of mass angle slip tends to be stable after 2.2s. The change in the center of mass angle slip of the SUV is stable until about 3.4s, but after 3.4s, the center of mass angle slip of the SUV increases sharply and stops after 4.1s, indicating that the vehicle has rolled over. Fig 11(b) shows the changes in the yaw angular velocity. The yaw angular velocity increases decreases and then increases in the opposite direction. After 2.2s, the yaw angular velocity of the SUV without control continues to increase and then decreases; the change is extremely unstable and finally stops at 3.4s. The yaw rate of the SUV without predictive control tends to be flat after 2.2s but does not stabilize and stops at around 4.1s. The yaw rate of the SUV with predictive control tends to be stable after 2.2s. The SUV's yaw angular velocity tends to stabilize. Fig 11(c) shows the evolution of the body roll angle. The body roll angle of the SUV without control and without predictive control increased sharply after 2.2s and stopped at 3.4s and 4.1s, respectively. The body roll angle of the SUV with predictive control increased after 2.2s but stabilized after that. Fig 11(d) expresses the variation of the *LTR* of the SUV. Without control and without predictive control, the *LTR* of the vehicle increased to more than 1 and then stopped. With predictive control, the *LTR* of the SUV underwent a gradual stabilization process and was maintained at approximately 0.85.

The phase transition diagram for $\lambda$-$\beta$ and $\beta$-$\dot{\beta}$ are unfolded in Fig 12. As illustrated in Fig 12(a), the vehicle phase trajectory without predictive control is extremely unstable and divergent. The vehicle phase trajectory without predictive control converges slightly better than the vehicle phase trajectory without control but fails to converge. The vehicle phase trajectory with predictive control converges to a stable focus. Similarly, as demonstrated in Fig 11(a), the vehicle phase trajectories without control and without predictive control are unstable and do not converge. In contrast, the vehicle phase trajectories with predictive control converge to a stable focus, indicating that the designed vehicle ARCS with time delay supplementation produces a better effect on the time delay effect of the control system.

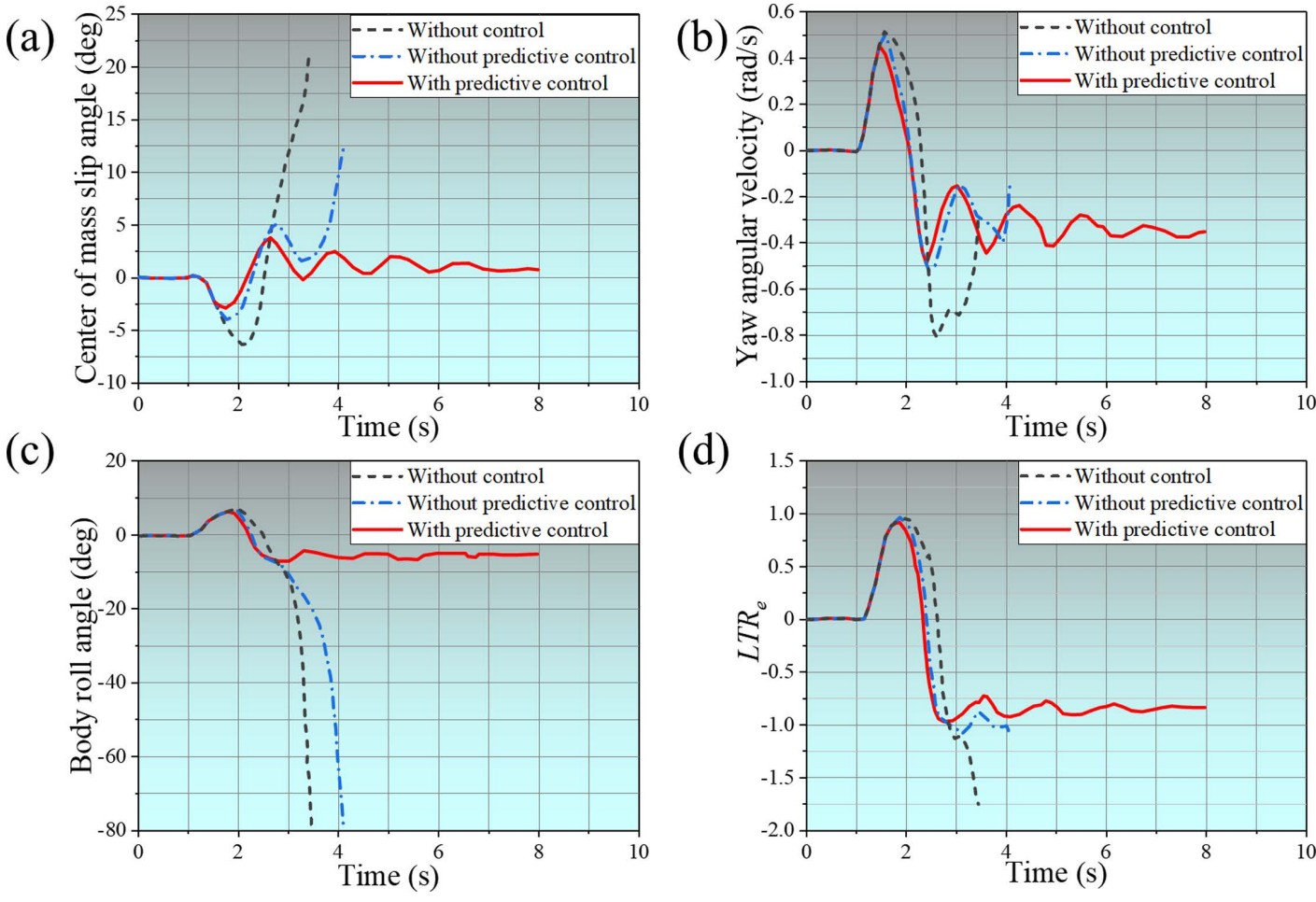

**Fig 11. Changes to Vehicle Rating Indicators.**

Combining Figs 11 and 12, it can be observed that in the fishhook steering condition, the vehicle rolled over at 3.4s without control and at 4.1s without predictive control. When the control is not applied, the vehicle exhibits obvious over-steer characteristics; the center of mass angle slip, yaw angle velocity, and body roll angle increase rapidly, the phase trajectory of the vehicle system is dispersed, and the SUV will roll over. In the case of not applying predictive control, although the addition of control can provide some control effect compared to not applying control, the presence of time delay makes control system's response too slow, and the vehicle eventually rolls over. After the prediction mechanism is introduced into the vehicle control system, the time delay effect caused by signal processing, feedback delay, mechanical response, and other factors is significantly improved. Due to the addition of the prediction mechanism, the system can predict the dynamic changes of the vehicle in advance and timely adjust the suspension system, braking force distribution, power output and other key parameters of the vehicle at the critical moment. Through this precise control, the vehicle's sideslip angle and body roll angle can be strictly controlled in a safe and stable area, and the *LTR* can also be maintained at a reasonable level of about 0.85. These ensure that the vehicle can maintain a good balance under lateral force and avoid insufficient tire grip or vehicle out of control caused by excessive load concentration. Finally, the vehicle remains stable throughout the driving process, successfully avoiding rollover accidents and improving driving safety and reliability.

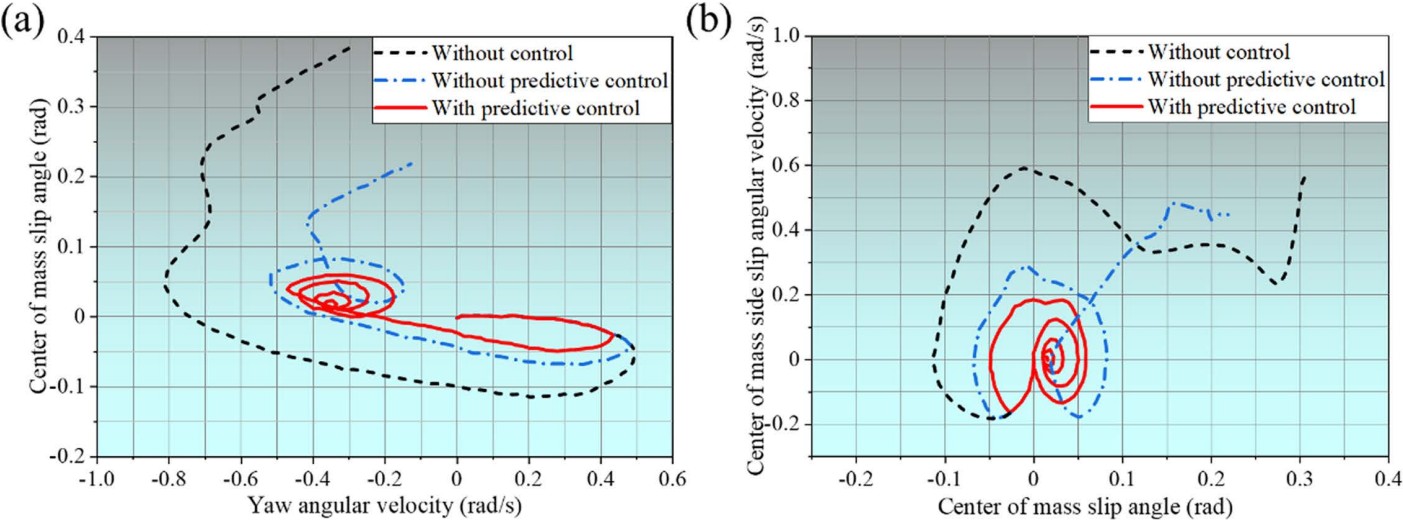

**Fig 12. Phase transition diagram.**

## 7. Conclusions

In order to prevent the SUV from rolling over when turning, this paper proposes a control system that can prevent the vehicle from rolling over. The system selects *LTR* as the rollover evaluation index. It uses the grey prediction model to predict the index in advance to compensate for the time delay in the system and to prevent the vehicle from rolling over in advance. The specific conclusions are as follows:

1) According to the characteristics of SUVs, a 3-DOF dynamic model of SUV rollover is established, and the *LTR* is used as the evaluation standard of vehicle rollover. Then, an *LTR* estimation model is established. The relative estimation error of the model is less than 2.7% through the J-turn test, which shows that the model can accurately estimate the *LTR* of SUVs.

2) Aiming at the time-delay problem of ARCS, the grey prediction model predicts the LTR of SUVs. The experimental data show that the maximum relative prediction error of the SUV *LTR* prediction model is less than 3%, which proves that this model can accurately predict the *LTR* of SUVs in advance.

3) Following the proposed active anti-rollover control method with time delay compensation, numerical simulations were performed under two different steering conditions, namely step and fishhook, and the results showed that the vehicle without control and without predictive control rolled over under both conditions. In the case of predictive control, the grey prediction system predicts in advance. It compensates for the time delay of the control system so that the center of mass slip angle of the vehicle can be controlled in the stable region, the *LTR* is maintained at about 0.85, and the vehicle does not roll over. To sum up, the method proposed in this paper has a good effect on compensating the time-delay phenomenon of ARCS. It can maintain the rollover dynamic stability of SUVs and improve the anti-rollover capability of SUVs.

In this paper, the co-simulation of CarSim and MATLAB/Simulink verifies that the proposed ARCS effectively avoids the rollover of SUVs during cornering in both step steering and fishhook steering conditions. However, the real-world vehicle cornering situations are complex and variable, and the system's actual performance has not been verified in the real world. Therefore, future work will focus on experimental testing to determine the reliability of the proposed ARCS in

real-world applications. For example, the control algorithm can be embedded in the prototype vehicle's electronic control unit (ECU), and the field experiment can be carried out in a controlled environment. In addition, the hardware in the loop (HIL) experiment can be used as an intermediate step to verify the real-time computing efficiency of the control system and the compatibility of automotive hardware. Integrating the system with automotive hardware and whether to compound safety standards are also tasks to be completed in the future.

## Abbreviations

| Symbol | Description | Unit | Symbol | Description | Unit |
|---|---|---|---|---|---|
| $a_f, a_r$ | front and rear wheel sideslip angles respectively | rad | $d$ | wheelbase | m |
| $F_{yfl}, F_{yfr}$ | lateral forces on left and right front SUV wheels | N | $l_f, l_r$ | distances from the center of gravity to front and rear axles respectively | m |
| $F_{yrl}, F_{yrr}$ | lateral forces on left and right rear SUV wheels | N | $\varphi$ | body roll angle | rad |
| $\delta$ | front wheel rotation angle | rad | $\varphi_r$ | slope angle of the road surface | rad |
| $\gamma$ | yaw angular velocity | rad/s | $h$ | distance from the center of gravity of the sprung mass to the roll center | m |
| $H$ | distance of the center of gravity of the unsprung mass from the road surface | m | $m$ | sprung mass | kg |
| $g$ | acceleration due to gravity | m/s² | $F_{zl}, F_{zr}$ | droop forces on the left and right wheels of the SUV | N |
| $\sum F_y$ | resultant lateral force on the SUV | N | $A_y$ | lateral acceleration | m/s² |
| $I_{xx}, I_{yy}, I_{zz}$ | moments of inertia of the sprung mass around the $X$, $Y$ and $Z$ axes respectively | kg·m² | $u, v$ | the lateral and longitudinal speeds of the SUVs, respectively | m/s |
| $\ddot{z}$ | vertical acceleration of the sprung mass | m/s² | $F_{xi}$ | the braking force of the brake wheel | N |
| $a$ | the distance from the center of mass to the front axle | m | $T_{bi}$ | the braking torque of the brake wheel | N·m |
| $R_i$ | the effective rolling radius of the tire | m | $P_{xi}$ | the wheel cylinder pressure of the brake wheel | pa |
| $C_i$ | the braking torque per unit wheel cylinder pressure of the brake wheel | N·m | | | |

## Author contributions

**Data curation:** Rong Wang.

**Methodology:** Dongtao Wang.

**Software:** Yifan Hu, Xuan Liu.

**Supervision:** Yanjian Shen.

**Validation:** Yifan Hu, Xuan Liu.

**Writing – original draft:** Dongtao Wang.

**Writing – review & editing:** Rong Wang.

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
