## [Decision Letter · Decision Letter 0]

18 Mar 2025

PONE-D-25-12675Research on anti-rollover active control of sports utility vehicle with time-delay compensation functionPLOS ONE

Dear Dr. Wang,

Thank you for submitting your manuscript to PLOS ONE. After careful consideration, we feel that it has merit but does not fully meet PLOS ONE’s publication criteria as it currently stands. Therefore, we invite you to submit a revised version of the manuscript that addresses the points raised during the review process.

We look forward to receiving your revised manuscript.

Kind regards,

Jinhao Liang

Academic Editor

PLOS ONE

“The Hunan Provincial Natural Science Foundation: 2023JJ60216 , the Hunan Provincial Department of Education Scientific Research Outstanding Youth Project: 24B1026 Recipient: Dongtao Wang. The Hunan Provincial Natural Science Foundation: 2023JJ60217 Recipient: Rong Wang.”

“This work was supported by the Hunan Provincial Natural Science Foundation (grant number 2023JJ60216, 2023JJ60217), the Hunan Provincial Department of Education Scientific Research Outstanding Youth Project (grant number 24B1026).”

“The Hunan Provincial Natural Science Foundation: 2023JJ60216 , the Hunan Provincial Department of Education Scientific Research Outstanding Youth Project: 24B1026 Recipient: Dongtao Wang. The Hunan Provincial Natural Science Foundation: 2023JJ60217 Recipient: Rong Wang.”

Reviewers' comments:

Reviewer's Responses to Questions

**Comments to the Author**

1. Is the manuscript technically sound, and do the data support the conclusions?

Reviewer #1: Yes

Reviewer #2: Partly

2. Has the statistical analysis been performed appropriately and rigorously? 

Reviewer #1: Yes

Reviewer #2: Yes

3. Have the authors made all data underlying the findings in their manuscript fully available?

Reviewer #1: Yes

Reviewer #2: Yes

4. Is the manuscript presented in an intelligible fashion and written in standard English?

Reviewer #1: Yes

Reviewer #2: No

5. Review Comments to the Author

Reviewer #1: Overall, the study is well-designed and demonstrates promising results. The work is well-documented and reproducible. Addressing the given points would further improve it for publication by making it more concise and readable, here are some review comments for the paper:

1. Discussion of how the approach could be validated further and integrated into practice scenarios.

2. The discussion covers the key strengths and limitations. Relating back to how it builds on and advances prior work could highlight the significance.

3. The background of the proposed research should be further explained in detail. Please note that the up-to-date of references will contribute to the up-to-date of your manuscript. In this sense, in Introduction, authors should consider the following relevant papers: Quantized Iterative Learning Control of Communication Constrained System with the Encoding and Decoding Mechanism, Transactions of the Institute of Measurement and Control; ADP-Based Prescribed-Time Control for Nonlinear Time-Varying Delay Systems With Uncertain Parameters, IEEE Transactions on Automation Science and Engineering; Spatiotemporal fault estimation for switched nonlinear reaction–diffusion systems via adaptive iterative learning, International Journal of Adaptive Control and Signal Processing; By this rigorous literature analysis, your research's original results and divergence from usual methods will be revealed, strengthening the context in which your research is placed.

4. How should the proposed method's features be tuned for optimal performance?

5. All references must be accurate, and complete. Verify all cited papers, add missing volume/issue/page information, correct errors, and provide DOIs for unpublished references, which still have no volume and issue.

6. The performance of the proposed method should be better analyzed, commented and visualized in the experimental section.

7. Have the authors considered employing other values in their research? What is the impact of the variables' sensitivity on the outcomes?

Reviewer #2: Abstract should be more about the problem studied, contributions and solutions, of course, in a short but informative way. Also, the abstract, like the title, should be able to stand alone and fully explain what your paper is about.

In the abstract and conclusion, the contribution of this paper is not well presented. In the conclusion and abstract, highlight the novelty of the paper.

The introduction is weak and should include the research question, the aim of the paper and the contribution.

In related work…. Many researches work on this idea. What is really the novelty as compared to other studies? What is the new and the difference between the previous works and present work?.

Improve the quality of literature along with the latest literature.

The explanation of the related work needs to be criticized and improved in general.

What about last updating in this topic and new references from 2019-2025? The survey of existing literature is not sufficient. It would useful to include in the Introduction of the paper some discussion on other possible real applications of the obtained results.

Figures are not clear. Clear diagrams and figures are required for readers to have clear images.

Improve the quality of figures for better visibility. It is blur that should be adjusted.

Weak conclusion.

Conclusion should be more specific with improvement writing quality.

A suggestion for future work should be added in the conclusion section.

- Rewrite the references according to journal template.

-Please strictly follow the instructions to the format specified in the journal template for preparing the paper

The format and English writing of this paper should be improved. The paper needs language revision.

6. PLOS authors have the option to publish the peer review history of their article (what does this mean? ). If published, this will include your full peer review and any attached files.

**Do you want your identity to be public for this peer review?** For information about this choice, including consent withdrawal, please see our Privacy Policy .

Reviewer #1: No

Reviewer #2: No

---

## [Author Response · Author response to Decision Letter 1]

16 Apr 2025

Dear editors and reviewers:

Thank you for your letter and for the reviewers' comments concerning our manuscript entitled “Research on anti-rollover active control of sports utility vehicle with time-delay compensation function”. The comments are all valuable and very helpful for revising and improving our paper, as well as the important guiding significance to our researches. We have studied comments carefully and have made correction which we hope meet with approval.

We have provided a point by point answer of each reviewer comment and highlighted (red letters) in the revised manuscript the passages that have been changed.

Reviewer 1

Overall, the study is well-designed and demonstrates promising results. The work is well-documented and reproducible. Addressing the given points would further improve it for publication by making it more concise and readable, here are some review comments for the paper

Response: Thank you for your affirmation of our article.

1. Discussion of how the approach could be validated further and integrated into practice scenarios.

Response: Thank you very much for your good comments. We apologise for not taking this into account. It's our fault. After revision, we added the supplementary content at the end of the conclusion and marked it red. For your convenient, the added content was also listed as follow:

In this paper, the co-simulation of CarSim and MATLAB/Simulink verifies that the proposed ARCS effectively avoids the rollover of SUVs during cornering in both step steering and fishhook steering conditions. However, the real-world vehicle cornering situations are complex and variable, and the system’s actual performance has not been verified in the real world. Therefore, future work will focus on experimental testing to determine the reliability of the proposed ARCS in real-world applications. For example, the control algorithm can be embedded in the prototype vehicle’s electronic control unit (ECU), and the field experiment can be carried out in a controlled environment. In addition, the hardware in the loop (HIL) experiment can be used as an intermediate step to verify the real-time computing efficiency of the control system and the compatibility of automotive hardware. Integrating the system with automotive hardware and whether to compound safety standards are also tasks to be completed in the future.

2. The discussion covers the key strengths and limitations. Relating back to how it builds on and advances prior work could highlight the significance.

Response: Thank you very much for your good comments. Your comments made us realize that there are indeed shortcomings in this area and also gave us great inspiration. According to your comments, we made a revision at the end of the introduction and marked it in red. For your convenient, the added content was also listed as follow:

To sum up, most of the existing researches on vehicle rollover prevention methods mainly focus on the control of body roll angle, which is not the best choice among the various indicators of vehicle rollover prevention and does not consider the impact of time delay on the system control performance. In order to solve the limitations of previous studies, first of all, in selecting vehicle anti-rollover indicators, this paper selects LTR, which is universal and intuitive, as the vehicle anti-rollover indicator. Then, it establishes and verifies the LTR estimation model of the SUV. Then, considering the time-delay problem in the control system, the LTR is predicted based on the grey prediction model. Finally, the fuzzy controller is controlled in advance according to the predicted LTR to ensure the control performance. Therefore, the control method proposed in this paper can compensate for the time delay of the vehicle’s active ARCS, ensure control accuracy, and improve the vehicle’s active safety.

3. The background of the proposed research should be further explained in detail. Please note that the up-to-date of references will contribute to the up-to-date of your manuscript. In this sense, in Introduction, authors should consider the following relevant papers: Quantized Iterative Learning Control of Communication Constrained System with the Encoding and Decoding Mechanism, Transactions of the Institute of Measurement and Control; ADP-Based Prescribed-Time Control for Nonlinear Time-Varying Delay Systems With Uncertain Parameters, IEEE Transactions on Automation Science and Engineering; Spatiotemporal fault estimation for switched nonlinear reaction–diffusion systems via adaptive iterative learning, International Journal of Adaptive Control and Signal Processing; By this rigorous literature analysis, your research's original results and divergence from usual methods will be revealed, strengthening the context in which your research is placed.:

Response: Thank you for the above three papers of great reference value, which have a great effect on the improvement of our research. We have cited these three papers to enrich our literature review.

For your convenient, the added content was also listed as follow:

Tao et al. studied a quantitative iterative learning control based on a coding and decoding mechanism, which effectively solved the problem of data loss in a grid control system when the communication bandwidth was limited, and the load was high [22].

Zhang et al. proposed a prescribed time adaptive dynamic programming control method, which can optimize the steady-state performance of the nonlinear time-varying delay system and ensure that the system can accurately complete the task within a specific time range [23].

Peng et al. designed a fault estimator for spatiotemporal faults and used the iterative learning strategy to estimate the faults accurately [24].

4. How should the proposed method's features be tuned for optimal performance?

Response: Thank you very much for your good comments. It is true that this problem is not well reflected in the article, which is our thoughtlessness. In the last part of Article 4.2 prediction of the LTR and analysis of its accuracy, we made a supplementary description and marked it in red. For your convenient, the added content was also listed as follow:

Through iterative experiments, this paper determines the optimal modeling dimension (m) and prediction step (k). Specifically, the higher modeling dimension improves prediction stability but reduces responsiveness to sudden system changes. On the contrary, a smaller prediction step (e.g., k=2) enhances the prediction accuracy but cannot fully compensate for the longer time delay. Therefore, the trade-off between accuracy and responsiveness is considered when selecting the best parameters. The most important thing about ARCS proposed in this paper is grey prediction, and the most important thing about grey prediction is the modeling dimension and prediction step. Selecting the best modeling dimension and prediction step can make the performance of ARCS the best.

5. All references must be accurate, and complete. Verify all cited papers, add missing volume/issue/page information, correct errors, and provide DOIs for unpublished references, which still have no volume and issue.

Response: Thank you very much for your good comments. I'm sorry that we didn't take this into account. We have modified the references quoted in this article according to the standard reference format. For your convenience, all the references cited are listed at the end of this responses.

6. The performance of the proposed method should be better analyzed, commented and visualized in the experimental section.

Response: Thank you very much for your good comments. We are very sorry that we have not done well in this regard. Your comments have been of great help to us. According to your comments, we have made supplementary explanations and marked in red in 6.1 step steering condition and 6.2 fishhook steering condition respectively. For your convenient, the added content was also listed as follow:

a. In the last paragraph of 6.1 step steering condition:

Comprehensive Fig 8 and Fig 9 show that under the step steering condition, the vehicle rollover time is about 3.6s when the vehicle is without control, and the vehicle rollover time is about 4.2s when the vehicle is without predictive control. The vehicle can complete the steering smoothly when the vehicle is with predictive control. When the vehicle turned without control, all evaluation indicators were unstable, indicating that the vehicle would reverse soon after turning. When the vehicle turned without predictive control, although all evaluation indicators were better than without control, the vehicle still eventually rolled over due to the time delay in the controller’s response. However, with predictive control, the grey prediction system can accurately predict the vehicle’s driving status and potential risks, pre-calculate the possible time delay problem, and take corresponding compensation measures in time. In this way, the vehicle will trigger the ARCS when it is steering, and the system can carry out the control operation quickly and effectively through advanced preparation and precise prediction. It can accurately adjust the distribution of the vehicle’s center of gravity, optimize the contact state between the tires and the ground, and ensure that the vehicle is always stable during the steering process, thus enabling the vehicle to safely and smoothly complete the steering action, effectively avoiding traffic accidents triggered by rollover and other dangerous situations, and safeguarding the safety of the vehicle and the passengers.

b. In the last paragraph of 6.2 fishhook steering condition:

Combining Fig 11 and 12, it can be observed that in the fishhook steering condition, the vehicle rolled over at 3.4s without control and at 4.1s without predictive control. When the control is not applied, the vehicle exhibits obvious oversteer characteristics; the center of mass angle slip, yaw angle velocity, and body roll angle increase rapidly, the phase trajectory of the vehicle system is dispersed, and the SUV will roll over. In the case of not applying predictive control, although the addition of control can provide some control effect compared to not applying control, the presence of time delay makes control system’s response too slow, and the vehicle eventually rolls over. After the prediction mechanism is introduced into the vehicle control system, the time delay effect caused by signal processing, feedback delay, mechanical response, and other factors is significantly improved. Due to the addition of the prediction mechanism, the system can predict the dynamic changes of the vehicle in advance and timely adjust the suspension system, braking force distribution, power output and other key parameters of the vehicle at the critical moment. Through this precise control, the vehicle’s sideslip angle and body roll angle can be strictly controlled in a safe and stable area, and the lateral load transfer rate can also be maintained at a reasonable level of about 0.85. These ensure that the vehicle can maintain a good balance under lateral force and avoid insufficient tire grip or vehicle out of control caused by excessive load concentration. Finally, the vehicle remains stable throughout the driving process, successfully avoiding rollover accidents and improving driving safety and reliability.

7. Have the authors considered employing other values in their research? What is the impact of the variables' sensitivity on the outcomes?

Response: Thank you very much for your good comments. I am very sorry that we did not take this into account in the article. According to your comments, we have revised and marked the text in red. For your convenient, the added content was also listed as follow:

a. In paragraph 2, line 10 of 4.2 Prediction of the LTR and analysis of its accuracy:

In order to analyze the prediction accuracy further, the low threshold may lead to the premature start of the system and increase energy consumption. A higher threshold may cause the system to respond too late and increase the risk of rollover. Through experimental analysis and comparison of the results of various data in 0.7~0.9, it can be selected that the control system takes the lateral load transfer rate equal to 0.85 as the control threshold, that is, when the lateral load transfer rate reaches 0.85, the ARCS of SUV will start to work.

b. In paragraph 3, line 5 of 4.2 Prediction of the LTR and analysis of its accuracy:

Since the difference between the predicted value and the estimated value of LTR with modeling dimension outside [4,10] is too large, the accuracy of the modeling dimension in [4,10] is only analyzed in detail below.

c. In paragraph 5 of 4.2 Prediction of the LTR and analysis of its accuracy:

The prediction and sampling time significantly impact the model accuracy, while the control threshold selection directly affects the system’s response time and safety. Therefore, the sensitivity of these variables should be carefully considered when designing and optimizing the ARCS. Changes in these variables will directly affect the ARCS’s performance.

Reviewer 2

1. Abstract should be more about the problem studied, contributions and solutions, of course, in a short but informative way. Also, the abstract, like the title, should be able to stand alone and fully explain what your paper is about.

Response: Thank you very much for your good comments. I am sorry that we have some deficiencies in the writing of the abstract of this article. Your comments are of great reference value to us. We have revised and marked red in the article. For your convenient, the revised content was also listed as follow:

The global incidence of traffic accidents caused by vehicle rollovers has exhibited a persistent upward trajectory in recent years. This paper proposes a novel rollover prevention control method incorporating time-delay compensation to address inherent latency issues in anti-rollover control systems (ARCS). First, structural parameters and dynamic theory establish a three-degree-of-freedom (3-DOF) dynamics model for a sport utility vehicle (SUV). Subsequently, a lateral load transfer ratio (LTR) estimation model is developed and validated under J-turn test conditions. A grey prediction model is then implemented to forecast LTR values in advance, compensating for system time delays. A two-dimensional fuzzy controller, utilizing error and error change rate as inputs, generates corrective yaw moment through differential braking to maintain vehicle stability. Co-simulation experiments conducted in CarSim and MATLAB/Simulink under typical driving scenarios demonstrate that the proposed method effectively mitigates ARCS time delays while preserving driving stability. The results suggest this approach provides both a practical solution for SUV rollover prevention and a conceptual advancement for vehicle active safety systems, showing strong potential for real-world implementation to reduce rollover risks and enhance road safety.

2. In the abstract and conclusion, the contribution of this paper is not well presented. In the conclusion and abstract, highlight the novelty of the paper.

Response: Thanks for your comments. Your comments have given us great inspiration in perfecting this article. According to your comments, we have revised and supplemented the article and marked it red. For your convenient, the modified content was also listed as follow:

a. In line 12 of the abstract, we add the contribution of the method proposed in this paper. As follows:

The results suggest this approach provides both a practical solution for SUV rollover prevention and a conceptual advancement for vehicle active safety systems, showing strong potential for real-world implementation to reduce rollover risks and enhance road safety.

b. In the conclusion part, we add the following content to reflect the novelty of the method proposed in this paper:

In order to prevent the SUV from rolling over when turning, this paper proposes a control system that can prevent the vehicle from rolling over. The system selects LTR as the rollover evaluation index. It uses the grey prediction model to predict the index in advance to compensate for the time delay in the sy

---

## [Decision Letter · Decision Letter 1]

21 Apr 2025

Research on anti-rollover active control of sports utility vehicle with time-delay compensation function

PONE-D-25-12675R1

Dear Dr. Wang,

We’re pleased to inform you that your manuscript has been judged scientifically suitable for publication and will be formally accepted for publication once it meets all outstanding technical requirements.

Kind regards,

Jinhao Liang

Academic Editor

PLOS ONE

Additional Editor Comments (optional):

The author has addressed the reviewers' comments well. It is recommended that the paper can be accepted for publication. Furthermore, the background on yaw moment control could refer to the recent work, such as "A Direct Yaw Moment Control Framework Through Robust T-S Fuzzy Approach Considering Vehicle Stability Margin, IEEE/ASME Transactions on Mechatronics, vol. 29, no. 1, pp. 166-178, Feb. 2024", and "A Robust Dynamic Game-Based Control Framework for Integrated Torque Vectoring and Active Front-Wheel Steering System, IEEE Transactions on Intelligent Transportation Systems, vol. 24, no. 7, pp. 7328-7341, July 2023".

Reviewers' comments:

Reviewer's Responses to Questions

**Comments to the Author**

1. If the authors have adequately addressed your comments raised in a previous round of review and you feel that this manuscript is now acceptable for publication, you may indicate that here to bypass the “Comments to the Author” section, enter your conflict of interest statement in the “Confidential to Editor” section, and submit your "Accept" recommendation.

Reviewer #1: All comments have been addressed

Reviewer #2: All comments have been addressed

2. Is the manuscript technically sound, and do the data support the conclusions?

Reviewer #1: Yes

Reviewer #2: Yes

3. Has the statistical analysis been performed appropriately and rigorously? 

Reviewer #1: Yes

Reviewer #2: Yes

4. Have the authors made all data underlying the findings in their manuscript fully available?

Reviewer #1: Yes

Reviewer #2: Yes

5. Is the manuscript presented in an intelligible fashion and written in standard English?

Reviewer #1: Yes

Reviewer #2: Yes

6. Review Comments to the Author

Reviewer #1: The topic appears interesting theoretically, and the applications are academic. There were some issues that have now been resolved. The results obtained are promising and accurate. The paper deserves to be published in this form.

Reviewer #2: I had a look at the revised manuscript PONE-D-25-12675R1, entitled "Research on anti-rollover active control of sports utility vehicle with time-delay compensation function". and checked whether the author(s) addressed all my comments/concern.

The revised paper has been revised according to my comments/concern.

My recommendation: Accepted without modifications.

7. PLOS authors have the option to publish the peer review history of their article (what does this mean? ). If published, this will include your full peer review and any attached files.

**Do you want your identity to be public for this peer review?** For information about this choice, including consent withdrawal, please see our Privacy Policy .

Reviewer #1: No

Reviewer #2: No

---

## [Editor Report · Acceptance letter]

PONE-D-25-12675R1

PLOS ONE

Dear Dr. Wang,

I'm pleased to inform you that your manuscript has been deemed suitable for publication in PLOS ONE. Congratulations! Your manuscript is now being handed over to our production team.

Kind regards,

on behalf of

Dr. Jinhao Liang

Academic Editor

PLOS ONE